# Integrative Analysis of the Nasal Microbiota and Serum Metabolites in Bovines with Respiratory Disease by 16S rRNA Sequencing and Gas Chromatography/Mass Selective Detector-Based Metabolomics

**DOI:** 10.3390/ijms231912028

**Published:** 2022-10-10

**Authors:** Ying Zhang, Chunji Ma, Yang Han, Hua Jin, Haixia Luo, Xiujing Hao, Min Li

**Affiliations:** 1Life Science School, Ningxia University, Yinchuan 750021, China; 2Key Laboratory of Ministry of Education for Conservation and Utilization of Special Biological Resources in the Western, Ningxia University, Yinchuan 750021, China

**Keywords:** bovine respiratory disease, nasal microbiota, metabolites

## Abstract

Bovine respiratory disease (BRD) continues to pose a serious threat to the cattle industry, resulting in substantial economic losses. As a multifactorial disease, pathogen infection and respiratory microbial imbalance are important causative factors in the occurrence and development of BRD. Integrative analyses of 16S rRNA sequencing and metabolomics allow comprehensive identification of the changes in microbiota and metabolism associated with BRD, making it possible to determine which pathogens are responsible for the disease and to develop new therapeutic strategies. In our study, 16S rRNA sequencing and metagenomic analysis were used to describe and compare the composition and diversity of nasal microbes in healthy cattle and cattle with BRD from different farms in Yinchuan, Ningxia, China. We found a significant difference in nasal microbial diversity between diseased and healthy bovines; notably, the relative abundance of *Mycoplasma bovis* and *Pasteurella* increased. This indicated that the composition of the microbial community had changed in diseased bovines compared with healthy ones. The data also strongly suggested that the reduced relative abundance of probiotics, including *Pasteurellales* and *Lactobacillales*, in diseased samples contributes to the susceptibility to bovine respiratory disease. Furthermore, serum metabolomic analysis showed altered concentrations of metabolites in BRD and that a significant decrease in lactic acid and sarcosine may impair the ability of bovines to generate energy and an immune response to pathogenic bacteria. Based on the correlation analysis between microbial diversity and the metabolome, lactic acid (2TMS) was positively correlated with *Gammaproteobacteria* and *Bacilli* and negatively correlated with Mollicutes. In summary, microbial communities and serum metabolites in BRD were characterized by integrative analysis. This study provides a reference for monitoring biomarkers of BRD, which will be critical for the prevention and treatment of BRD in the future.

## 1. Introduction

Bovine respiratory disease (BRD) is a serious health problem in cattle worldwide [1]. Bovines of all ages can be affected, but calves are most susceptible [2]. BRD leads to poor health, high morbidity and mortality, reduced carcass weight, impaired animal welfare, and increased treatment and vaccination costs in infected herds, which has brought substantial economic losses to the cattle breeding industry [3]. BRD is a complex disease and may be attributed to a combination of factors, including microbes, environmental conditions, age, condition, nutrition, duration of travel, and other stress such as weaning, air quality/particulates, and vaccination status. Among them, pathogen infection and respiratory microbial imbalance are important causative factors [4,5,6]. The viral pathogens most often identified are *Bovine herpesvirus type 1*, *Bovine parainfluenza-3*, *Bovine coronavirus*, and *Bovine viral diarrhea virus* [7,8], while common bacterial pathogens include *Mannheimia haemolytica*, *Pasteurella multocida*, *Histophilus*, and *Mycoplasma* [9,10]. Increasing evidence suggests the importance of pathogens encountering the resident microbiota and other pathogens while colonizing the host. Pathogen–host interactions can affect BRD pathogenesis, including increased bacterial adhesion, enhanced host defense, and modulation of the immune response of one microbe to the benefit of another [11,12]. It has been reported that the nasopharynx and trachea of healthy feedlot cattle and those with bacterial bronchopneumonia had distinct bacterial metacommunities, and that in healthy steers, L. lactis and L. casei may provide colonization resistance against bacterial respiratory pathogens [13]. Thus, the resident microbiota in the nasopharynx has positive implications for preventing respiratory pathogens from establishing an infection [6,14].

Although colonization of the nasopharynx by pathogens and changes in metabolites in the serum of the organism are the precursors to BRD [15], little information is available on the relationship between nasal microbiota and serum metabolites that occur within this niche. Moreover, it is unclear whether there are regional differences in the nasal microbial communities of cattle. In our study, we collected nasal swabs and serum samples from bovines from five large-scale cattle farms in Ningxia, China. Then, we used a combination of high-throughput sequencing and Gas Chromatography/Mass Selective Detector (GC-MSD)-based metabolomics intending to understand the interactions between pathogens and serum metabolites, as well as relating microbial communities and changes in metabolites to the risk of developing BRD. Our results provide new insights into variations in nasal microbial communities and serum metabolites to better understand the etiology of BRD, which is critical for improving animal health.

## 2. Results

### 2.1. Alpha-Diversity Analysis Comparing Healthy Bovines and Those with Respiratory Disease

Following amplification and sequencing of 16S rRNA from the deep nasal swabs, the raw data were cleaned by checking by FASTQC software and filtered to remove adaptor or low-quality sequences. The data volume of clean reads was distributed between 35,615 and 78,264, and the effective sequences obtained for analysis were allocated between 15,577–45,590 (Appendix A). Valid tags were divided into operational taxonomic units (OTUs) according to 97% similarity by QIIME software, Appendix A shown these OTU clusters and an annotation of each sample. As shown in Figure 1A, the number of OTUs in the cattle with respiratory disease (D group) was significantly higher than that in the healthy (H) group, and a total of 8369 OTUs were shared between the two groups. As seen in Figure 1B–G, there was a higher Simpson index in nasal swabs of the H group compared with the D group, but a greater Chao 1 index in the D group compared with the H group, which shows the differences and diversity between the two groups. The values of various alpha-diversity indexes including the Shannon diversity index, Simpson diversity index, phylogenetic diversity (PD)_whole_tree, Goods_coverage index, Chao 1, and observed_species richness are shown in Appendix A. Taken together, these indices revealed that the sequencing data were adequate and that, compared with the H group, bovines in the D group had richer bacterial communities, but with a reduced evenness.

### 2.2. Characterization of the Nasal Microbial Communities in Healthy and Diseased Cattle

According to OTU annotations, the top nine dominant flora of the D and H groups at the phylum, class, order, family, and genus levels are shown in Figure 2. In both the H and D groups, at the phylum level, the top two species with the highest abundance were *Proteobacteria* and *Firmicutes*. The third most prominent phylum in the D group was *Tenericutes*, while that in the H group was *Bacteroidetes* (Figure 2A). At the class taxon, the top two in both the D group and the H group were *Gammaproteobacteria*, *Bacilli*, while *Mollicutes* were third in the D group and *Clostridia* was the third in the H group (Figure 2B). The most prominent orders identified in the D group were *Pseudomonadales*, *Pasteurellales*, and *Mycoplasmatales*. However, in the H group, *Pseudomonadales*, *Pasteurellales*, and *Lactobacillales* were most abundant (Figure 2C). Significant differences were also observed at the family level, where three significantly changed families were subordinated to three significantly changed orders. The relative abundances (RAs) of *Moraxellaceae* and *Pasteurellaceae* were greatly enhanced both in the D and H groups. The RAs of *Mycoplasmataceae* were significantly increased in the D group compared with the H group (Figure 2D). At the genus level, although the RAs of *Moraxella*, *Enhydrobacter*, *Moraxellaceae-other*, and *Aggregatibacter* were as prominent in the D and H groups, the RAs of *Mycoplasma* and *Pasteurella* were more prominent in the D group compared with the H group (Figure 2E).

To further demonstrate the microbiota structure of each sample, we created a heatmap of RAs at the phylum level (Figure 2F). In summary, the RAs of phyla in the D group were significantly (*p* < 0.05) different from those in the H group. In the four taxons, i.e., phylum, class, order, and family, the RAs of both the D group and the H group were ranked third. The genus *Mycoplasma* was the top microflora that allowed us to distinguish the D group from the H group, although *Pasteurella* was also abundant in the D group.

### 2.3. The Diversity and Composition of Nasal Microbiota Are Significantly Altered in Cattle with BRD

We initially assessed the diversity and richness of the bovine nasal microbial community through alpha-diversity analysis and RA display of bacterial composition. To better understand the role of microbiota diversity, the Unweighted-UniFrac and Weighted-UniFrac algorithms were used to measure the sample distance and analyze the beta-diversity (Figure 3A,B). Unweighted-UniFrac analysis demonstrated a significant difference in nasal microbial diversity between groups D and H (*p* < 0.05) (Figure 3A). At the same time, we constructed an evolutionary tree to display the similarity of whole samples (Appendix A). Then, we performed a principal coordinate analysis (PCoA) (Appendix A) and principal component analysis (PCA) (Appendix A), as well as partial least-squares discriminant analysis (PLS-DA) (Appendix A) on the OTU classification information of each sample. There was evident clustering separation among OTUs, revealing the distinct community structures between the two groups. In addition, a multiple response permutation procedure (MRPP) analysis also revealed that the bacterial community structures of the D group and the H group were significantly different (*p* <0.01) (Appendix A). Collectively, these data confirmed that the composition of the microbial community had changed in diseased bovines compared with healthy ones.

To confirm which type of bacteria was altered in diseased cattle affected by BRD, we performed high-dimensional class comparisons using the linear discriminant analysis of effect size (LEfSe). According to Figure 3C,D, *Mycoplasmataceae* was the key type of bacteria in group D, while *Firmicutes* was prominent in group H. A Metastats analysis (Appendix A) and t-test analysis (Appendix A) reconfirmed this result. In order to clarify which *mycoplasma* in the *Mycoplasma* genus was a significantly different species, we performed metagenomic sequencing on the above samples and visualized the species annotation by KRONA. As shown in Appendix A, *Mycoplasma bovis* was the key species allowing us to make a distinction between diseased and healthy cattle.

### 2.4. Evaluation of Serum Metabolites Reveals Alterations in the Metabolome of BRD Cattle

To explore the metabolic changes in diseased cattle, a metabolomic analysis was performed on serum from diseased and healthy cattle. We identified a total of 41 metabolites. Appendix A shows the sample IDs, and the quantitative results of each metabolite are shown in Appendix A. Then, to visualize the differences among the metabolite data, we performed PCA (Appendix A) and PL-SDA (Appendix A) analyses and evaluated the model by cross-validation (Appendix A) and permutation tests (Appendix A). Both the score and loading plots showed higher agglomeration and reproducibility among these samples, and the two groups exhibited different metabolite profiles. As shown in Appendix A, multiple metabolites, including lactic acid (8), urea (68), and hexadecanoic acid (182), made important contributions to the ability to discriminate between the samples. Table 1 presents the Align ID and the top 15 differential metabolites in the serum samples. Furthermore, the variable importance in projection (VIP) scores were utilized to quantify the metabolites that affected the difference between the biofluids from the two groups. As shown in Appendix A, the levels of lactic acid (8), phosphoric acid (83), and sarcosine (11) in the samples from the D group were significantly lower than those in samples from the H group. Statistical analysis using the t-test (Appendix A) also indicated that the differences in metabolism of lactic acid (8) and phosphoric acid (83) were equally significant (*p* < 0.05). In contrast, compared with samples from the H group, the metabolisms of urea (68) and hexadecanoic acid (182) showed higher levels in samples from the D group, although their VIP score was <1.

### 2.5. Correlations between Nasal Microbes and Serum Metabolites

To integrate the available data, we studied the putative correlations between species abundances and serum metabolites for the two groups (D and H), clustering those bacteria with a similar metabolomic profile in heatmap plots. As shown in Figure 4A, clear clusters were formed between microbiota and metabolites. For example, lactic acid (2TMS) showed a positive correlation with *Gammaproteobacteria* and *Bacilli* while showing a negative correlation with *Mollicutes*. Interestingly, we found that non-dominant bacterial species also had strong associations with other non-dominant metabolites. For example, *Fibrobacteria*, *Betaproteobacteria*, and *Ellin6529* showed a strong correlation with glycerol. Furthermore, to determine the relationship between metabolites and microbial gene functions, we used the same heatmap to display an association between the 41 metabolites screened from metabolomics and microbial gene functions (Figure 4B). The analyses showed that cardiovascular disease was the most relevant biological process associated with the metabolite phosphoric acid, although the neurodegenerative disease and the immune system also showed a positive correlation with phosphoric acid. However, the most represented metabolite, lactic acid, was positively correlated to most gene functions, including cardiovascular disease, cancer, and the digestive system.

## 3. Discussion

Evidence is accumulating for a potential role of the nasal microbiota in the pathogenesis of cattle with respiratory disease [16,17]. Using 16S rRNA amplicon and metagenomic sequencing, we investigated nasal microbial characteristics in BRD cattle and found that these subjects carry an altered nasal microbiota. *Mollicutes*, *Actinobacteria*, and *Flavobacteria*, which were of low prevalence in healthy cattle, were abundantly enriched in diseased subjects. Recent studies in southern Alberta, northwest Piedmont, and Egypt all demonstrated that *Mycoplasma*, a member of *Mollicutes*, was one of the most enriched genera in the microbiota of the upper (nasal swab) and lower (tracheal aspiration) respiratory tracts of bovines with respiratory disease [9,13,18,19]. Our results are in agreement with the literature, as we identified *Mycoplasma* as one of the genera with high relative abundance among bovines with respiratory disease in Ningxia, China. Metagenomic sequencing further confirmed that *Mycoplasma bovis* is one of the main pathogenic bacteria that can cause respiratory disease in cattle. This finding also suggests there are no regional differences in the effects of *Mycoplasma bovis* on bovine respiratory disease. It can be seen that members of the genus Mycoplasma, which are believed to play a negative role in the respiratory tract microbiota, participate in the development and progression of respiratory disease.

Differently, our study also observed a higher alpha- and beta-diversity in nasal microbiota of cattle with respiratory disease, indicating a more heterogeneous community structure among diseased cattle than among healthy cattle. At the class level, although *Gammaproteobacteria* and *Bacilli* were the top two dominant flora in both groups, the enrichment with *Bacilli* was significantly lower in the diseased group compared with the healthy group. Also, members of the *Clostridia*, which was the third most abundant in healthy cattle, decreased in the diseased group. The RAs and changes of the most abundant taxa at subordinated levels were further compared and we get a similar trend. *Lactobacillales* belong to the Bacilli Class [20]. Our study showed that nasal microbiota in the healthy group displayed a predominance of the *Pasteurellales*, and *Lactobacillales*, which were decreased in the diseased group. It is well known that some members of the *Pasteurellales* and *Lactobacillales* have positive significance in resisting pathogenic infection [21,22]. Several species of the *Lactobacillales* such as lactic acid bacteria (LAB) have potential as topical probiotics in the respiratory tract [23,24,25]. Since Lactobacillus strains were most active in inhibiting the adherence of *M. haemolytica* to bovine turbinate cells, they were selected as the best candidates for intranasal bacterial therapeutics to mitigate *M. haemolytica* infection in cattle [26]. Notably, Lactobacillus plantarum DR7, isolated from bovine milk, protected cattle against upper respiratory tract infections [27]. The bacteriocins produced by several members of the *Streptococcus* genus could act as “natural” antibiotics to treat or prevent bacterial infections [28]. Evidently, a reduced abundance of members of *Pasteurellales* and *Lactobacillales* may also contribute to the susceptibility of bovines to respiratory disease. This further hints at a correlation between the pathogenesis of respiratory disease and microbiome homeostasis.

In order to determine whether changes in bacterial communities had an impact on metabolic output, the metabolic changes in serum samples were evaluated by GC-MSD-based metabolomics. In the present study, we found that bovines with respiratory disease had lower serum concentrations of lactic acid, phosphoric acid, and sarcosine than healthy cattle. Lactate, as a fulcrum of metabolism, has the potential to reshape the energy metabolism of an organism [29,30]. Lactate is involved in multiple metabolic pathways, including glycolysis, the pentose phosphate pathway, and the oxidative metabolism of pyruvate [31,32,33]. It plays a key role in energy regulation, immune tolerance, and cellular DNA damage repair processes [34,35]. There is compelling evidence that glucose and glycogen catabolism may proceed to lactate production under fully aerobic conditions [36]. Previously, lactic acid was identified as a metabolite of great importance in classifying animals as BRD or non-BRD [37]. On the other hand, sarcosine is an endogenous amino acid involved in one-carbon metabolism [38]. Sarcosine is downregulated suggesting respiratory disease effect one-carbon metabolism uptake. Taken together, we speculate that reduced levels of lactate and sarcosine imply slower glucose and carbon metabolism, causing affected individuals to have a diminished ability to mount an immune response to pathogens due to having reduced metabolic reserves. While the mechanisms remain unclear, these data indicate that lactic acid and sarcosine play a key role in the maintenance of the homeostasis of nasopharyngeal microbiota and that reduced levels of lactate are possibly related to increased instances of disease. In contrast, there was an upward trend in urea and hexadecanoic acid levels in diseased cattle, although the changes failed to reach statistical significance. We speculate that this may be related to individual differences in cattle and an insufficient sample size, but that the specific reasons need to be explored.

An integrative analysis of the nasal microbiome and serum metabolome was performed to gain insights into the links between nasal flora dysbiosis and respiratory disease and we found that lactic acid (2TMS) levels in serum were positively correlated with *Gammaproteobacteria* and *Bacilli*, but negatively correlated with *Mollicutes*. In our study, bovines with respiratory disease had a lower abundance of *Lactobacillales* and decreased lactic acid levels, but an increased abundance of *Mycoplasma bovis*, the main pathogen. This demonstrates the potential probiotic properties of LAB strains and confirms the hazards of *Mycoplasma bovis* to cattle. In addition, from our analysis of the association of microorganisms and metabolites, we found that non-dominant microorganisms also had strong associations with non-dominant metabolites. BRD is a multifactorial disease and a mixed infection by multiple pathogenic microorganisms can lead to metabolic disorders of organisms [8]. Due to the diversity and complexity of the microbial ecosystem, we only analyzed dominant microbes and metabolites at present. Whether other non-dominant microbes may also provide an important contribution to BRD remains to be elucidated and requires further studies.

## 4. Materials and Methods

### 4.1. Specimens

From February until May 2019, bovine nose swab samples and serum samples were collected from five large breeding farms in Ningxia, China (N 38°200, E 106°160). A health check of the herd was performed prior to collecting samples. A total of 72 deep nasal swabs from 18 bovines with respiratory disease were collected using sterile disposable swab collectors (4 replicates were collected from each animal). At the same time, a total of 44 deep nasal swabs from 11 healthy bovines were collected. Furthermore, serum was collected from 15 diseased bovines and 10 healthy bovines from the same farms that were injected with anesthetics. All specimens were sealed and placed on ice, and their information was registered in detail. They were transported to the Key Laboratory of the Ministry of Education for Protection and Utilization of Special Biological Resources in western China within 2 h where the plasma was separated by centrifugation and stored at −80 °C until further use. The study protocol was reviewed and approved by the Animal Welfare & Ethics Committee of Ningxia University (Ningxia, China) (No. NXU-2019-008).

### 4.2. 16S rRNA Sequencing

Total DNA from deep nasal swab samples was extracted using the QIAamp DNA Mini Kit (QIAGEN, Hilden, Germany) following the manufacturer’s instructions. Next, the bacterial 16S rRNA V3-V4 region was selected for PCR amplification. Then, the PCR products were purified using the QIAquick Gel Extraction Kit (QIAGEN, Hilden, Germany), and a NanoDrop or Qubit were used to quantify the library. According to the data volume of 50,000 reads per sample, an appropriate library volume was added, and HiSeq2500/MiSeq was selected for PE250 sequencing.

### 4.3. DNA Extraction and Metagenome Sequencing

Total DNA extracted from nasal swabs was quantified using a NanoDrop (Thermo Scientific, Waltham, MA, USA) and confirmed by 0.8% agarose gel electrophoresis. The DNA was broken into DNA fragments with 350 bp approximately. A library based on the samples was constructed following end repair, sequencing adaptor addition, and purification. Each library was sequenced using Illumina HiSeq 2000 equipment and the key species that distinguish diseased from healthy bovines were identified by metagenomic sequencing.

### 4.4. Microbial Composition and Diversity Analysis

Based on the OTU clustering and annotation of the samples, bacterial alpha-diversity was determined by the R package “QIIME2”, and presented by observed Shannon, Simpson, PD_whole_tree, Goods_coverage, Chao 1, and Observed_species indices. Sequence RAs were calculated for every sample and each taxon, and constructed the matrix on bacterial phylum, class, order, family, and genus level, respectively. PCoA and PCA were used to display the samples’ microflora beta-diversity. Unweighted-Unifrac distances and weighted-Unifrac distances were calculated using the QIMME package. Based on the normalized relative abundance matrix, using the non-parametric factorial Kruskal-Wallis (KW) sum-rank test to estimate the features with significantly different RAs (*p* < 0.05). Finally, Linear discriminant analysis (LDA) was used to assess the effect size of each feature (logLDA score = 3.6 as cutoff).

### 4.5. Metabolomics Processing

An aliquot (20 μL) of each serum sample was accurately transferred to a 1.5 mL Eppendorf tube. Next, 1 mL extracting solution (isopropanol: acetonitrile: water = 3:2:2) and 5 μL myristic acid solution (3 mg/mL) were added and the mixture was vortexed for 1 min, followed by centrifugation (13,000 rpm) at 4 °C for 15 min to obtain the supernatant, which was then dried under nitrogen. Subsequently, following these derivatization steps, 20 μL of methoxyamine hydrochloride/pyridine (40 mg/mL) were added and incubated at 30 °C for 90 min. Next, 90 μL of N-Methyl-N-(trimethylsilyl) trifluoroacetamide containing 1% Trimethylchlorosilane were added and incubated at 37 °C for 30 min. The derivatized samples were centrifuged (13,000 rpm) at 4 °C for 5 min and the supernatant was used for LC-MS analysis. An equal volume of the prepared samples was mixed and divided into 7 quality control samples to monitor the precision and stability of the instrument. For data processing, GC-MSD metabolic profiling was performed using Agilent 7890A and Agilent 5975C MSDs. The chromatographic column used was Agilent DB5-MS (30 m × 250 mm × 0.25 μm), the inlet temperature was 250 °C, and the MSD interface temperature was 230 °C. The flow rate was set to 1.1 mL/min, and the injection volume was 1 μL. Data acquisition was done in full-scan mode (m/z ranged from 50 to 500).

### 4.6. Statistical Analysis of Metabolomics

The raw GC-MSD data were converted to a NetCDF format using Agilent Chrom Station software (Agilent Technologies, Santa Clara, CA, USA). Under the R software platform, the eRah procedure was used for peak identification, retention time alignment, automatic integration, and other preprocessing. Then, we compared the spectral data with the Golm Metabolome database (GMD) to select the results with higher retention similarity and retention index and obtained qualitative results including retention time, retention index, metabolite name, and CAS number. Furthermore, PCA and PLS-DA were used to observe the overall metabolic differences between the D and H groups, and cross-validation (CV) was used to evaluate the quality of the PLS-DA model. The VIP of PLS-DA was employed to measure the importance and contribution of metabolites. Meanwhile, a two-tailed Student’s t-test of normalized peak areas was also used to assess the statistical significance. Metabolites with VIP values > 1.0 and *p* < 0.05 were considered differential metabolites.

### 4.7. Correlation Profiling between the Metabolites and Nasal Microbiotas

A Pearson correlation analysis was performed between altered metabolites screened from serum metabolomics and perturbed nasal microbial genera screened from 16S rRNA gene sequencing. The screen criteria included a correlation coefficient |r| > 0.60, *p* < 0.05. Finally, significantly correlated metabolites and nasal microbial classes were obtained and displayed with heatmaps, where blue indicated positive correlation and red indicated negative correlation.

## 5. Conclusions

In conclusion, the current study found that the altered levels of several metabolites, especially lactic acid, phosphoric acid, and sarcosine, were associated with microbial changes, indicating an abnormality in metabolism and microbial levels in bovines with respiratory disease. The microbes identified from the nasal swabs, such as *Mycoplasma bovis*, *Pasteurellales* and *Lactobacillales*, provide new insights into the pathogenesis of BRD. The metabolites identified in serum might play various roles in supporting the needs of bovines for energy purposes and potential unidentified immune responses. Those indexes may serve as potential biomarkers for monitoring BRD prior to the appearance of clinical signs. Regulating the composition and stability of bacterial microbiota in the nasopharynx could prevent the onset of bovine respiratory disease and play a major role in the prevention and control of BRD.

## Figures and Tables

**Figure 1 ijms-23-12028-f001:**
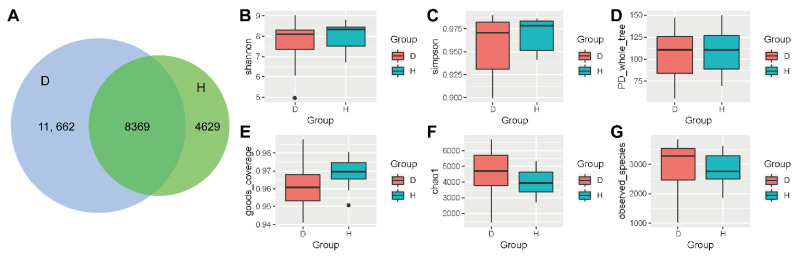
Alpha-diversity indexes between healthy cattle and cattle with respiratory disease. (**A**) OTUs of the whole samples. (**B**–**G**) Boxplots of alpha-diversity indexes comparing the D group and the H group. (**B**) Shannon diversity index. (**C**) Simpson diversity index. (**D**) phylogenetic diversity (PD_whole_tree). (**E**) goods_coverage. (**F**) Chao 1. (**G**) Observed_species. D: bovines with respiratory disease, H: healthy bovines.

**Figure 2 ijms-23-12028-f002:**
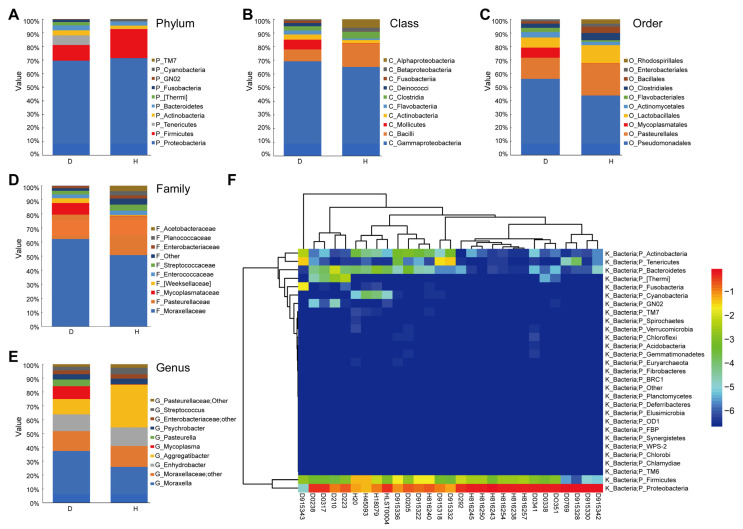
Relative abundance and taxonomic composition in bovine nasal microbial samples. (**A**) Phylum. (**B**) Class. (**C**) Order. (**D**) Family. (**E**) Genus. (**F**) Heat map analysis of the composition of the most abundant microbiota at the phylum level. Horizontal clustering indicates that the abundance of the species in different samples is similar; vertical clustering indicates similar expression levels of all species in different samples. Shorter branch length is associated with a greater similarity. D: bovines with respiratory disease, H: healthy bovines.

**Figure 3 ijms-23-12028-f003:**
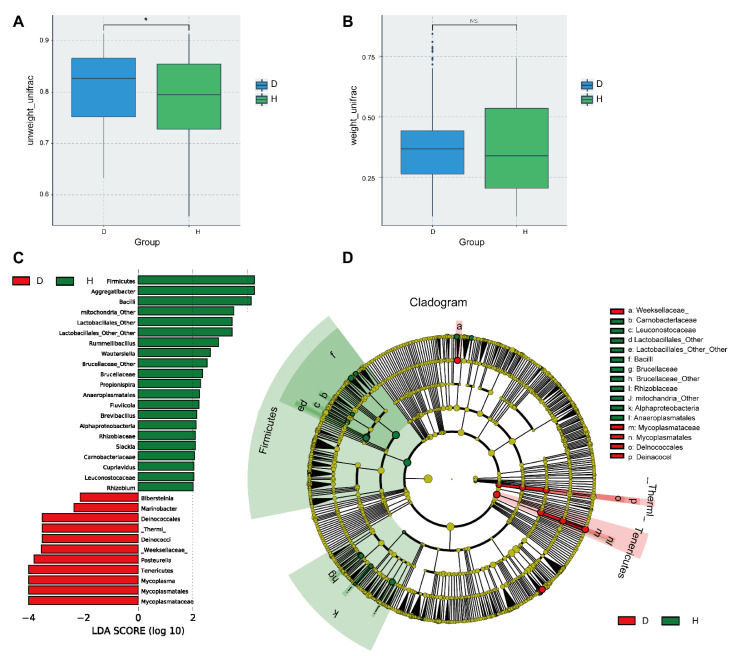
Beta-diversity and LEfSE analysis. Boxplots of the beta-diversity of nasal microbiota comparing the diseased (D) group and the healthy (H) group. The comparison between groups were performed by T-test. * *p* < 0.05, significant, NS, not significant. (**A**) Unweighted-UniFrac. (**B**) Weighted-UniFrac. (**C**) The LDA score was computed from differences in feature abundance between the D and H groups. Features were selected according to logLDA score > 3.6. (**D**) Taxonomic cladogram from LEfSe, depicting the taxonomic association between microbial communities from the D and H groups. Each node represents a specific taxonomic type. Yellow nodes denote taxonomic features that are not significantly different between the D and H groups. Red nodes denote the taxonomic types with greater abundance in the D group, while green nodes represent the taxonomic types that are more abundant in the H group. D: bovines with respiratory disease, H: healthy bovines.

**Figure 4 ijms-23-12028-f004:**
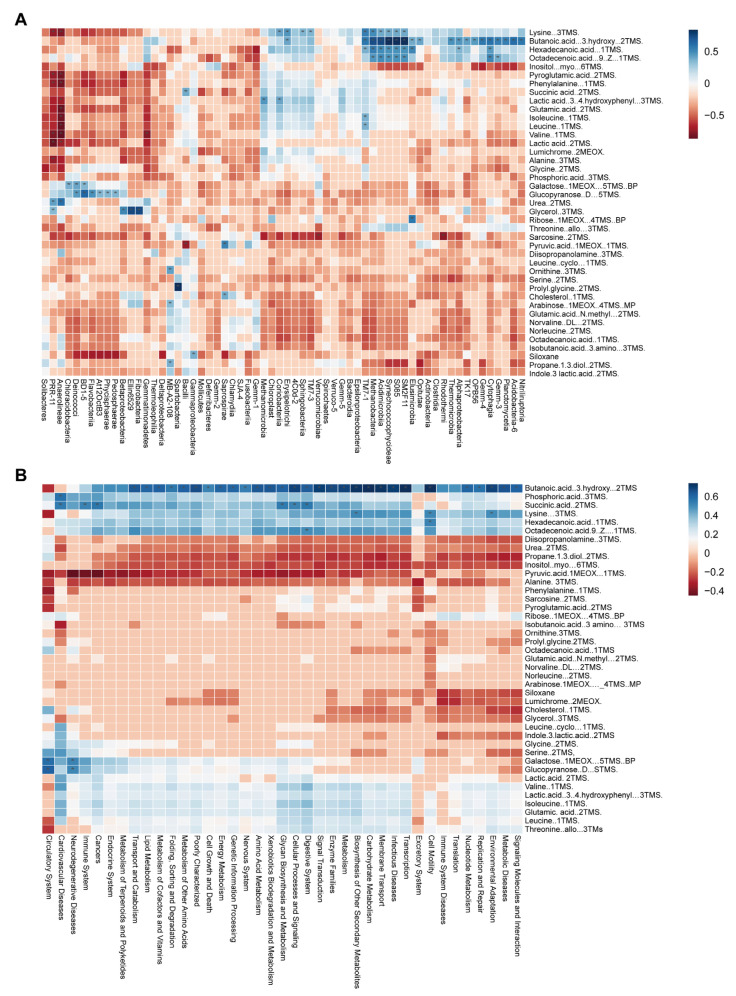
Correlation of nasal microbiota and serum metabolites in cattle. (**A**) Heat map summarizing the correlation between nasal flora and serum metabolites. (**B**) Heat map summarizing the correlation between gene function and metabolites. The red and blue gradient color bar visually shows the correlation between microbes and metabolites. A darker color is associated with a stronger correlation. Red indicates a positive correlation and blue indicates a negative correlation.

**Table 1 ijms-23-12028-t001:** The top 15 metabolites in serum by gas chromatography mass spectrometry detector.

Align ID	Name	Synon
8	Lactic acid (2TMS)	(S)-Lactic acid
83	Phosphoric acid (3TMS)	[PO(OH)3]
11	Sarcosine (2TMS)	(methylamino)acetic acid
10	Valine (1TMS)	(2S)-2-amino-3-methylbutanoic acid
63	Norvaline, DL-(2TMS)	(2S)-2-aminopentanoic acid
130	Pyroglutamic acid (2TMS)	(S)-(-)-2-Pyrrolidone-5-carboxylic acid
68	Urea (2TMS)	Carbamide
27	Leucine (1TMS)	2-amino-4-methylpentanoic acid
95	Norleucine (2TMS)	(2S)-2-aminohexanoic acid
131	Glutamic acid (2TMS)	(2S)-2-aminopentanedioic acid
51	Leucine, cyclo-(1TMS)	1-Amino-1-cyclopentanecarboxylic acid
52	Isoleucine (1TMS)	Hile
170	Lactic acid, 3-(4-hydroxyphenyl)-(3TMS)	2-hydroxy-3-(4-hydroxyphenyl) propanoic acid
182	Hexadecanoic acid (1TMS)	—
122	Glutamic acid, N-methyl-(2TMS)	(2S)-2-(methylamino) pentanedioic acid

## Data Availability

The original contributions presented in the study are included in the article/Appendix A.

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
