# Peer review of "Integrative Analysis of the Nasal Microbiota and Serum Metabolites in Bovines with Respiratory Disease by 16S rRNA Sequencing and Gas Chromatography/Mass Selective Detector-Based Metabolomics"

_ijms, 2022, doi:10.3390/ijms231912028_

Round 1
Reviewer 1 Report
1. Overall: The use of 16S rRNA sequencing, analysis of microbial diversity of bovine respiratory tract and comparison with elements of metabolism is of interest to a broad audience and especially to producers and scientists of feedlot cattle and dairy calves. There are broader implications of the research for new approaches to the prevention, diagnosis and management of BRD.
2. Abstract:
a. Line 13-14. ” Bovine respiratory disease (BRD) continues to pose a serious threat to the cattle industry, 13 resulting in immeasurable economic losses.” The economic losses in Yinchuan might not have been measured but losses caused by BRD have been measured in other countries. Suggest substituting “significant” or “substantial” instead of “immeasurable.”
b. Lines 14-15. “Both pathogen infection and respiratory microbial imbalance can cause BRD.” BRD complex is so called because it is a complex disease involving microbes, environmental conditions, age, condition, nutrition, duration of travel, other stressor such as weaning, air quality/particulates and vaccination status of calves. These factors are nicely summarized in the Chai et al., 2022 reference.
c. Lines 20-22. “We found that Mycoplasma bovis and Pasteurella were highly abundant in all diseased samples, while known bacterial pathogens associated with BRD including Haemophilus and Mannheimia haemolytica were not detected with high relative abundance in most samples.” M. bovis, Pasteurella spp., Haemophilus somnii and M. haemolytica are all associated with BRD and are also commensals.
d. Line 23. “This indicated that Mycoplasma bovis and Pasteurella were the main bacterial pathogens that caused BRD.” Finding M. bovis and Pasteurella spp. In nasopharyngeal samples does not establish causality for BRD as these are commensal bacteria, i.e., part of the normal flora of the bovine upper respiratory tract as indicated by your data.
e. Line 104. Define RA.
f. Fig. 5A & B. The labels on the heat maps are illegible.
Author Response
Dear Reviewers,
Thank you for your time and your comments for our manuscript of “Integrative analysis of the nasal microbiota and serum metabolites in bovines with respiratory disease by 16S rRNA sequencing and gas chromatography/mass selective detector-based metabolomics” We have taken all these comments and suggestions into account, and have made modifies in the uploaded revised manuscript for your review.
Comments:
- Overall: The use of 16S rRNA sequencing, analysis of microbial diversity of bovine respiratory tract and comparison with elements of metabolism is of interest to a broad audience and especially to producers and scientists of feedlot cattle and dairy calves. There are broader implications of the research for new approaches to the prevention, diagnosis and management of BRD.
- Abstract:
- Line 13-14. “Bovine respiratory disease (BRD) continues to pose a serious threat to the cattle industry, 13 resulting in immeasurable economic losses.” The economic losses in Yinchuan might not have been measured but losses caused by BRD have been measured in other countries. Suggest substituting “significant” or “substantial” instead of “immeasurable.”
- Lines 14-15. “Both pathogen infection and respiratory microbial imbalance can cause BRD.” BRD complex is so called because it is a complex disease involving microbes, environmental conditions, age, condition, nutrition, duration of travel, other stressor such as weaning, air quality/particulates and vaccination status of calves. These factors are nicely summarized in the Chai et al., 2022 reference.
- Lines 20-22. “We found that Mycoplasma bovis and Pasteurella were highly abundant in all diseased samples, while known bacterial pathogens associated with BRD including Haemophilus and Mannheimia haemolytica were not detected with high relative abundance in most samples.” M. bovis, Pasteurella spp., Haemophilus somnii and M. haemolytica are all associated with BRD and are also commensals.
- Line 23. “This indicated that Mycoplasma bovis and Pasteurella were the main bacterial pathogens that caused BRD.” Finding M. bovis and Pasteurella spp. In nasopharyngeal samples does not establish causality for BRD as these are commensal bacteria, i.e., part of the normal flora of the bovine upper respiratory tract as indicated by your data.
- Line 104. Define RA.
- Fig. 5A & B. The labels on the heat maps are illegible.
We also appreciate your clear and detailed feedback and hope that the explanation has fully addressed all of your concerns. In the remainder of this letter, we discuss each of your comments individually along with our corresponding responses.
Point 1: Line 13-14. “Bovine respiratory disease (BRD) continues to pose a serious threat to the cattle industry, 13 resulting in immeasurable economic losses.” The economic losses in Yinchuan might not have been measured but losses caused by BRD have been measured in other countries. Suggest substituting “significant” or “substantial” instead of “immeasurable.”
Response 1: We substituting “substantial” instead of “immeasurable.” The following is the modified content, which marked in yellow. (Lines 13-14 of abstract and lines 41-42 of introduction.)
Point 2: Lines 14-15. “Both pathogen infection and respiratory microbial imbalance can cause BRD.” BRD complex is so called because it is a complex disease involving microbes, environmental conditions, age, condition, nutrition, duration of travel, other stressor such as weaning, air quality/particulates and vaccination status of calves. These factors are nicely summarized in the Chai et al., 2022 reference.
Response 2: We have read the Chai et al., 2022 reference, modified the description in abstract and supplemented the detailed factors that can cause BRD in introduction. The modified content marked in yellow is the supplemented content. (Lines 14-16 of abstract and 42-46 of introduction.)
Point 3: Lines 20-22. “We found that Mycoplasma bovis and Pasteurella were highly abundant in all diseased samples, while known bacterial pathogens associated with BRD including Haemophilus and Mannheimia haemolytica were not detected with high relative abundance in most samples.” M. bovis, Pasteurella spp., Haemophilus somnii and M. haemolytica are all associated with BRD and are also commensals.
Response 3: We have recognized the error in the description, and we have modified this description in revised manuscript and marked in yellow. (Lines 21-23 of abstract.)
Point 4: Line 23. “This indicated that Mycoplasma bovis and Pasteurella were the main bacterial pathogens that caused BRD.” Finding M. bovis and Pasteurella spp. In nasopharyngeal samples does not establish causality for BRD as these are commensal bacteria, i.e., part of the normal flora of the bovine upper respiratory tract as indicated by your data.
Response 4: We agree with your point very much. After consideration, we modified this description in revised manuscript and marked in yellow. (Lines 23-24 of abstract.)
Point 5: Line 104. Define RA.
Response 5: We have defined the RA in manuscript. (Lines 108 of abstract.)
Point 6: Fig. 5A & B. The labels on the heat maps are illegible.
Response 6: We have modified the resolution of Figure 5 and uploaded the original image in Supplementary Material, hoping to see the labels clearly.
We would like to take this opportunity to thank you for all your time involved and this great opportunity for us to improve the manuscript. We hope you will find this revised version satisfactory.
Sincerely,
Ying Zhang

Reviewer 2 Report
The Zhang’s paper provides a reference for monitoring biomarkers of BRD using Integrative analysis of 16S rRNA sequencing and metabolomics.
This study allowed comprehensive identification of the changes in microbiota and metabolism associated with BRD, under standing which pathogens are responsible for the disease, and developing new therapeutic strategies.
I found the paper very interesting and I think it combined two innovative techniques. I think it is a complex one and I would propose to move figures 3, 4 and 5 into the additional materials.
Author Response
Dear Reviewers,
Thank you for your time and your comments for our manuscript of “Integrative analysis of the nasal microbiota and serum metabolites in bovines with respiratory disease by 16S rRNA sequencing and gas chromatography/mass selective detector-based metabolomics” We have taken all these comments and suggestions into account, and have made modifies in the uploaded revised manuscript for your review.
Comments:
The Zhang’s paper provides a reference for monitoring biomarkers of BRD using Integrative analysis of 16S rRNA sequencing and metabolomics.
This study allowed comprehensive identification of the changes in microbiota and metabolism associated with BRD, understanding which pathogens are responsible for the disease, and developing new therapeutic strategies.
I found the paper very interesting and I think it combined two innovative techniques. I think it is a complex one and I would propose to move figures 3, 4 and 5 into the additional materials.
We also appreciate your clear and detailed feedback and hope that the explanation has fully addressed all of your concerns. In the remainder of this letter, we discuss your comments with our corresponding responses.
Point 1: I think it is a complex one and I would propose to move figures 3, 4 and 5 into the additional materials.
Response 1: Thank you for your suggestion. As suggestion, we have moved Figure 4 into the additional materials, and modified the figures number. In manuscript, figure 3 show the beta-diversity and LEfSE analysis between healthy cattle and cattle with respiratory disease, and figure 5 display the correlation of nasal microbiota and serum metabolites in cattle. These figures are the core of the results. So, I'm sorry to cannot move Figure 3 and Figure 5 into the additional materials. (lines 166, 169, 174-175, and 177 of result 2.4)
We would like to take this opportunity to thank you for all your time involved and this great opportunity for us to improve the manuscript. We hope you will find this revised version satisfactory.
Sincerely,
Ying Zhang
